# Occurrence of Antimicrobial Resistance Genes in the Oral Cavity of Cats with Chronic Gingivostomatitis

**DOI:** 10.3390/ani11123589

**Published:** 2021-12-18

**Authors:** Wayne Tsang, Annika Linde, Janina A. Krumbeck, Guangxi Wu, Young J. Kim, Gerald H. Lushington, Tonatiuh Melgarejo

**Affiliations:** 1College of Veterinary Medicine, Western University of Health Sciences, Pomona, CA 91766, USA; wtsang@westernu.edu (W.T.); alinde@westernu.edu (A.L.); yjkim@westernu.edu (Y.J.K.); 2MiDOG LLC, 14672 Bentley Cir, Tustin, CA 92780, USA; jkrumbeck@zymoresearch.com (J.A.K.); gwu@zymoresearch.com (G.W.); 3Qnapsyn Biosciences, Inc., Lawrence, KS 66044, USA; glushington@qnapsyn.com

**Keywords:** feline chronic gingivostomatitis, microbiome, mycobiome, antibiotics, antimicrobial resistance, *Malassezia restricta*

## Abstract

**Simple Summary:**

Oral disease in cats is a significant concern in the small animal practice setting. The specific cause of this disease is inadequately understood; however, antibiotics are commonly used for the management, although many cats respond poorly to these treatments. Antibiotics have been overused and misused in the context of both human and veterinary medicine. Consequently, many antimicrobial drugs are becoming less effective in treating infections. This study aimed to evaluate the presence of antimicrobial resistance genes in swabs obtained from the mouth of cats. Moreover, the study looked at simultaneous occurrence between these genes and one type of oral fungi. We found that antimicrobial resistance genes are extremely common in both clinically healthy and sick cats. Furthermore, we established that Malassezia (a type of fungi) co-occurs with some resistance genes. The findings are important because antimicrobial resistance genes present in the mouth of cats have the potential to transfer to humans and thereby make certain antibiotics less effective.

**Abstract:**

Feline chronic gingivostomatitis (FCGS) is a severe immune-mediated inflammatory disease with concurrent oral dysbiosis (bacterial and fungal). Broad-spectrum antibiotics are used empirically in FCGS. Still, neither the occurrence of antimicrobial-resistant (AMR) bacteria nor potential patterns of co-occurrence between AMR genes and fungi have been documented in FCGS. This study explored the differential occurrence of AMR genes and the co-occurrence of AMR genes with oral fungal species. Briefly, 14 clinically healthy (CH) cats and 14 cats with FCGS were included. Using a sterile swab, oral tissue surfaces were sampled and submitted for 16S rRNA and ITS-2 next-generation DNA sequencing. Microbial DNA was analyzed using a proprietary curated database targeting AMR genes found in bacterial pathogens. The co-occurrence of AMR genes and fungi was tested using point biserial correlation. A total of 21 and 23 different AMR genes were detected in CH and FCGS cats, respectively. A comparison of AMR-gene frequencies between groups revealed statistically significant differences in the occurrence of genes conferring resistance to aminoglycosides (ant4Ib), beta-lactam (mecA), and macrolides (mphD and mphC). Two AMR genes (mecA and mphD) showed statistically significant co-occurrence with *Malassezia restricta*. In conclusion, resistance to clinically relevant antibiotics, such as beta-lactams and macrolides, is a significant cause for concern in the context of both feline and human medicine.

## 1. Introduction

Feline Chronic Gingivostomatitis (FCGS) is a severe immune-mediated inflammatory disease of the oral cavity in domestic cats [1]. Initial bacterial biofilm formation leading to plaque deposits results in activation of the immune system and gingivitis, which impacts individual teeth with gradual spread to surrounding tissues [2]. Despite substantial research efforts dedicated to the exploration of disease causation in FCGS, the etiology has yet to be established. The syndrome is likely multi-factorial, including the involvement of different pathogens, nutritional factors, environmental stressors, and more. Conventional treatment strategies to address FCGS involve medical management through the prescription of immunosuppressants and antibiotics or surgical management aimed at partial or complete dental extraction dependent on disease severity [3]. Given the variation in how patients with FCGS respond to immunosuppressants, efforts have increasingly focused on differentiating between the relative importance of bacterial pathogens, hereunder anaerobic and Gram-negative bacteria, such as *Pasteurella multocida* [1]. Broad-spectrum antibiotics are frequently prescribed empirically in cases with FCGS in the absence of culture and sensitivity analysis, which is a One Medicine (i.e., intersection between human, animal, and environmental health) issue when viewed in light of growing antimicrobial resistance concerns worldwide.

While antibiotics are routinely administered in an attempt to combat feline oral pathogens, our team has demonstrated that the oral cavity of patients with FCGS also harbors fungal species, such as *Malassezia restricta,* that are virtually absent in clinically healthy cats [4]. The coexistence of bacterial and fungal communities might suggest that fungi play a role in FCGS. The existing literatures suggest Malassezia species are pathogenic in human gut inflammation and some cancers [5]. Conversely, Malassezia species have been documented to serve prominent roles as commensal microorganisms in the oral cavity of humans [6]. In companion animal medicine, Malassezia species have been implicated in dermatitis, but no association has been reported to oral disease in either cats or dogs. Culture-independent analyses confirm that the oral cavity in cats is polymicrobial [7]. Metagenomic analyses of the feline oral mycobiome still remain scarce, and reports have thus far focused on cats with allergies versus controls [8]. A single report exists on the oral mycobiome in FCGS patients [4]. This study presents patterns of antimicrobial resistance (AMR) genes in cats with FCGS versus controls, as well as the co-occurrence of AMR genes with one highly abundant oral fungal species.

## 2. Materials and Methods

### 2.1. Subjects Included and Study Design

A total of 28 samples were analyzed in this study, 14 from clinically healthy (CH) cats and 14 from cats with chronic gingivostomatitis (FCGS). The CH samples were all collected at Western University of Health Sciences (WUHS), Pet Health Center (PHC), Pomona, CA, USA (*n* = 14). The FCGS samples were obtained from WUHS-PHC (Pomona, CA, USA), Saddleback Animal Hospital (Tustin, CA, USA), Advanced Veterinary Specialty Group (Tustin, CA, USA), and Cat Care Clinic (Orange, CA, USA). Six different cat breeds were represented (Domestic Short Hair, 17; Domestic Long Hair, 8; Siamese, 1; Scottish Fold, 1; Main Coon, 1; Persian, 1) across a wide age range (2 to 13 years old) with an average of 4 ± 0.5 years in the CH group and 8 ± 1.1 years in the FCGS group. All cats from the FCGS group were diagnosed with chronic gingivostomatitis ≥6 months prior to the inclusion in our study. Co-existing viral infection status and previous medical/surgical therapy information (e.g., tooth extractions and antibiotic therapy) were not available at the time of the oral sample collection. The CH cats were recruited from the surgery section of the WUHS-PHC at the time of admission for elective procedures (e.g., neutering). All cats in the CH and FCGS groups underwent the same examination protocol and were classified based on a previously published FCGS scale with minor modifications [9]. Briefly, the oral cavity was examined thoroughly, and lesions were sorted according to severity, as previously described [4]: grade 0, absence of lesions; grade 1, mild gingivitis; grade 2, moderate gingivitis; grade 3, severe gingivitis; grade 4, gingivitis associated with proliferative and/or ulcerative lesions in the caudal oral cavity/palatoglossal fold and/or alveolar, labial, buccal, sublingual, and lingual mucosae (extra-gingival lesions). Only cats with grade 4 lesions were included in the FCGS group. Cats in the CH group presented no evidence of oral lesions or only mild gingivitis (grade 1 in six of 14 cats) at the time of examination.

### 2.2. Sample Collection, DNA Extraction, Preparation, and Sequencing

Oral samples from the FCGS cats were collected in mucosal transition areas (affected tissues and their contiguous normal areas) using a sterile DNA-free swab (HydraFlock^®^, Puritan^®^ Cat. No. 25-3406-H, Guilford, ME, USA). For the CH cats, swab samples were collected from the gingival, hard palate, rostral dorsal tongue, and other oral mucosal surfaces. All samples were immediately immersed and preserved in DNA/RNA Shield^TM^ (Zymo Research Corp.; Cat. No. R1108, Irvine, CA, USA) until processing at the MiDOG LLC testing facility (Tustin, CA, USA). Genomic DNA was purified using the ZymoBIOMICSTM-96 DNA kit (Cat. No. D4304, Zymo Research Corp., Irvine, CA, USA). Sample library preparation and data analysis for bacterial and fungal profiling were performed by Zymo Research Corp. using the Quick-16S NGS Library Prep Kit (Cat. No. D6400, Zymo Research Corp., Irvine, CA, USA) with minor modifications. Primer sequences are proprietary to the MiDOG LLC service and target the 16S rDNA V1–V3 region for bacteria and the ITS-2 region for fungal analysis. Libraries were sequenced using an Illumina HiSeq 1500 sequencer, and reads were filtered through Dada2 (R package version 3.4) [10]. Phylotypes were computed as percent proportions based on the total number of sequences in each sample. The relative abundances of bacteria compared to fungi were determined, assuming an equivalency of one 16S rDNA copy to one fungal ITS copy. Species-level resolution of the sequencing approach used here has previously been demonstrated by shot-gun sequencing [11].

### 2.3. Detection of AMR Genes, and Correlation Analysis with Malassezia restricta

The presence of AMR genes in the oral cavity of the study subjects was evaluated by using a proprietary sequencing workflow that targets at least eighty AMR genes. An amplicon based sequencing approach was applied using proprietary PCR primers, which were designed based on AMR gene sequences retrieved from NCBI (National Center for Biotechnology Information). Sequencing reads were mapped back to the reference using a proprietary pipeline. To ensure specificity and reproducibility of the tests, sequencing reads were further confirmed using the Comprehensive Antibiotic Resistance Database (CARD) [12]. Patterns in the occurrence of AMR genes (proportions) between groups were compared using two-sample z-tests (*p* < 0.05). Point-biserial correlation tests (stats v3.6.1 R Core Team, R Foundation for Statistical Computing, Vienna, Austria, 2013) were applied to analyze any relationship between the presence of specific AMR gene (dichotomous variable) and relative abundance of *Malassezia restricta* (continuous variable), a species found solely in oral swabs from FCGS cats based on prior microbial core analysis [4].

## 3. Results

AMR genes were detected in all 28 samples regardless of feline health status (Figure 1). A total of twenty-four acquired AMR genes were detected with the potential to confer resistance to a wide range of antimicrobials, including clinically relevant and commonly used antibiotic classes such as beta-lactams, tetracyclines, aminoglycosides, phenicols, lincosamides, macrolides and sulfonamides. A total of 21 and 23 different AMR genes were detected in CH and FCGS cats, respectively (Table 1). Statistically significant differences were found in occurrence of genes conferring resistance to aminoglycosides (ant4Ib, *p* = 0.012), beta-lactam (mecA, *p* = 0.005), and macrolides (mphD, *p* = 0.014 and mphC, *p* = 0.024). These four AMR genes were found in 7, 8, 5, and 10 cats with FCGS, as compared to 1, 1, 0, and 4 CH controls, respectively (Table 1).

Additionally, we established co-occurrence between *Malassezia restricta*, the most frequently found fungal species in the oral cavity of FCGS cats [4], with two of the twenty-four AMR genes. Specifically, we found a positive correlation between the abundance of *M. restricta* and the presence of mecA (r = 0.49, *p* = 0.0076) and mphD (r = 0.62, *p* = 0.0004) genes (Table 2).

## 4. Discussion

This is the first study to investigate patterns of AMR gene expression in cats with FCGS versus CH controls. Twenty-four acquired AMR genes were identified in the oral cavity of all 28 cats in this study, and at least one AMR gene was detected in each sample. Compared to CH controls, AMR genes conferring resistance to aminoglycosides, beta-lactams, and macrolide antibiotics were found more commonly in oral swabs from cats with FCGS.

The mecA (beta-lactam) AMR gene is of considerable clinical relevance because it confers resistance to commonly prescribed antibiotics used in the management of FCGS, including amoxicillin/clavulanate and cephalexin [13]. The study includes a relatively small sample size, but the data give pause for thought when deciding on the empirical use of amoxicillin/clavulanate in the management of FCGS. The cautious use of amoxicillin/clavulanate (Clavamox) is especially important because this is one of the most common antibiotics used for the treatment of infections in feline medicine, such as FURC (feline upper respiratory complex) and feline eosinophilic plaques and lip ulcers [14]. Our findings moreover have One Medicine implications because beta-lactams, such as amoxicillin/clavulanate (Augmentin), and cephalexin (Keflex) are among the top five most commonly prescribed outpatient antibiotics in the USA [15]. Furthermore, the mecA gene is expressed by multidrug-resistant bacteria, such as methicillin-resistant *Staphylococcus aureus* (MRSA). MRSA is among the most important causes of nosocomial outbreaks and is responsible for significant morbidity and mortality around the world [16]. Further research is needed to explore the public health significance of mecA and other AMR genes in domestic cats, hereunder the transmission of AMR genes to humans (Figure 2).

An additional two AMR genes were found more commonly in the oral cavity of cats with FCGS compared to CH controls, including mphD and mphC, which confer resistance to macrolide antibiotics such as azithromycin and clarithromycin. Azithromycin has been suggested as an alternative antibiotic in refractory cases of FCGS but has shown only limited success. The rationale behind the use of azithromycin is that some non-responsive FCGS cases present with positive *Bartonella henselae* titers. Interestingly, pradofloxacin (a third-generation quinolone antibiotic used in companion animals) has superior in vitro antimicrobial activity against *B. henselae* [17]. Azithromycin has been used for more than a decade in feline medicine as a treatment for Bartonellosis, empirical treatment of FURC in shelter cats [18], and acute cytauxzoonosis (infection with tick-borne haemoprotozoan parasite) [19]. Public health concerns about the presence of this AMR gene in client-owned cats has not been investigated, but it warrants serious consideration since azithromycin is the second most prescribed outpatient antibiotic for humans in the USA [20] and a critically important antibiotic for human medicine according to the WHO [21].

The AMR tetW/N/W gene was ubiquitously present in the oral cavity of all cats enrolled in this study (Figure 1). This gene confers resistance to tetracycline and all its synthetic derivatives that inhibit bacterial aminotransferase-tRNA [12]. Doxycycline (a synthetic tetracycline derivative) has been one of the most commonly used antibiotics for FCGS but has shown limited success in changing clinical outcomes [13]. According to the Centers for Disease Control and Prevention [20], doxycycline was prescribed 19.5 million times in 2020, making it the fifth most used oral antimicrobial in outpatients in the USA. The presence of the tetW/N/W gene in all 28 cats in this study raises important public health questions about the potential of cat-to-human AMR gene transfer and the original source of this gene in the cat population.

The presence of a single AMR gene that confers resistance to aminoglycosides (ant4Ib) in affected cats appears to have less veterinary medical relevance considering that cats with chronic gingivostomatitis rarely are treated with this class of antibiotics. The use of aminoglycosides in cats is typically avoided because of well-established adverse effects such as nephrotoxicity and ototoxicity [22].

Our group recently documented the oral microbiome and mycobiome of CH and FCGS cats [4], which showed that the oral cavity of cats with FCGS harbors *Malassezia restricta*—a pathogen commonly associated with seborrheic dermatitis in humans [23]—at a significantly higher abundance and frequency as compared to CH cats. Notably, the composition of cell wall polysaccharides of *M. restricta* is distinctly different from all other fungal species analyzed to date [24], suggesting that *M. restricta* has evolved unique traits to adapt more effectively to the skin and mucosal host microenvironments. Moreover, *M restricta* expresses a metallo-beta-lactamase [25] capable of catalyzing the hydrolysis of all beta-lactam antibiotic classes, including carbapenems [26]. This constitutes a serious health concern because there is currently no clinically relevant inhibitor available for the fungal metallo-beta-lactamases [26]. In light of these observations, we decided to explore the co-occurrence of this fungal species and clinically relevant AMR genes in the oral cavity of FCGS cats and established a significant association of *M. restricta* with two highly relevant AMR genes (mecA and mphD) in the oral cavity of cats with FCGS. Although the co-occurrence between *M. restricta* and select AMR genes does not offer proof of causation based on this study alone, it opens the door for further investigating the role that *M. restricta* may play in a feline oral health context.

In summary, the AMR genes documented in this descriptive study have the potential to confer antimicrobial resistance to the five most frequently prescribed outpatient oral antibiotics in 2020 in the USA, including amoxicillin, azithromycin, amoxicillin/clavulanic acid, cephalexin, and doxycycline [20].

## 5. Conclusions

The feline oral resistome, as described here, includes at least 24 acquired AMR genes with potential for horizontal gene transfer to the bacteriome of humans and other animals. We also found the occurrence of four AMR genes to be significantly different between FCGS and CH cats, where genes conferring resistance to critically important antibiotics, including aminoglycosides, beta-lactam, and macrolides, were more commonly found in cats with FCGS. The study furthermore showed the co-occurrence between *M. restricta* (a beta-lactamase producing yeast) and two AMR genes in FCGS cats, strongly suggesting that scientifically based manipulation of the oral microbiome (bacterial and fungal) may prove extremely helpful in designing novel and effective strategies to manage a feline syndrome that has been a medical conundrum for small animal clinicians for decades. Finally, our findings also have important One Medicine implications that warrant further investigation of the public health significance of resistomes in cats.

## Figures and Tables

**Figure 1 animals-11-03589-f001:**
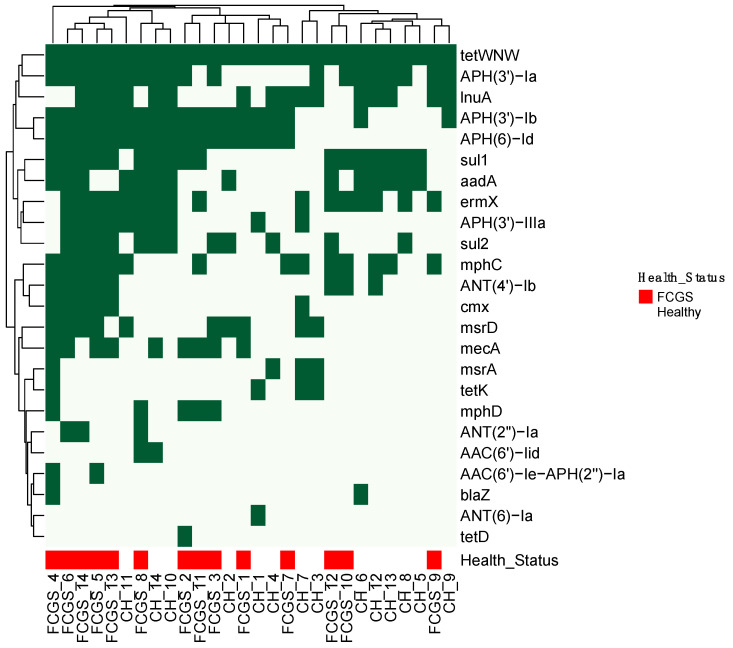
AMR gene presence in feline oral swab samples. Dark green indicates a gene is present, while light green indicates the absence of the gene. Heatmap was built using R (hierarchical clustering with Euclidean distance, complete-linkage). Samples and AMR genes were grouped by similarity. Health status is shown as red (feline chronic gingivostomatitis, FCGS) or white (clinically healthy).

**Figure 2 animals-11-03589-f002:**
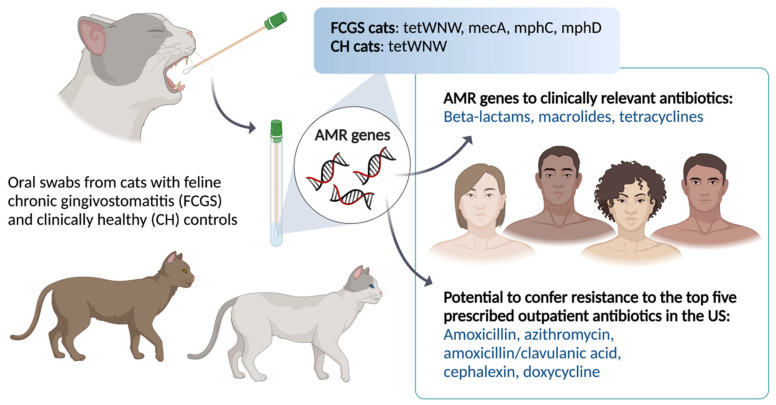
Acquired AMR genes in clinically healthy and diseased cats may spillover into human populations and thus represent a less known public health concern. Created with BioRender.com.

**Table 1 animals-11-03589-t001:** AMR gene presence in the number of oral samples from clinically healthy (CH, *n* = 14) cats and cats with feline chronic gingivostomatitis (FCGS, *n* = 14). The two sample proportions were compared using a z-score test.

AMR Gene	Antibiotic Resistance	CH	FCGS	z-Score	*p*-Value
aph33Ib	Aminoglycosides	8	11	−1.22	0.22
aph3Ia	Aminoglycosides	9	10	−0.40	0.69
aph6Id	Aminoglycosides	6	11	−1.93	0.05
ant2Ia	Aminoglycosides	0	3	−1.83	0.07
ant4Ib	Aminoglycosides	1	7	−2.51	0.01 *
aac6IId	Aminoglycosides	1	1	0.00	1.00
aac6Ie	Aminoglycosides	0	2	−1.47	0.14
aadA	Aminoglycosides	9	5	1.51	0.13
ant6Ia	Aminoglycosides	1	0	1.02	0.31
aph3IIIa	Aminoglycosides	5	5	0.00	1.00
mecA	Beta-Lactams	1	8	−2.83	0.00 *
blaZ	Beta-Lactams	1	1	0.00	1.00
cmx	Florfenicols	1	5	−1.85	0.06
sul1	Sulfonamides	7	10	−1.16	0.25
sul2	Sulfonamides	6	7	−0.76	0.45
mphD	Macrolides	0	5	−2.47	0.01 *
mphC	Macrolides	4	10	2.26	0.02 *
lnuA	Lincosamides	10	6	1.52	0.13
tetD	Tetracyclines	0	1	−1.02	0.31
tetWNW	Tetracyclines	14	14	0.00	1.00
tetK	Tetracyclines	3	1	1.08	0.28
ermX	Macrolides, Lincosamides	7	9	−0.76	0.45
msrA	Macrolides, Tetracyclines, Lincosamides	3	1	1.08	0.28
msrD	Macrolides, Tetracyclines, Lincosamides	4	6	−0.79	0.43

* Statistically significant values (*p* < 0.05).

**Table 2 animals-11-03589-t002:** Co-occurrence of *Malassezia restricta* and select AMR genes.

AMR Gene	Correlation Coefficient	*p*-Value
mecA	0.49381	0.00757 *
mphD	0.62317	0.0004 *
mphC	−0.07606	0.70048
Ant4lb	0.08136	0.68066

* Statistically significant values (*p* < 0.05).

## Data Availability

Third Party Data Restrictions apply to the availability of these data. Data was obtained from MiDOG LLC and the data includes proprietary information regarding the sequencing specifics. Data for the abundance tables are available from the corresponding author (T Melgarejo) upon reasonable request.

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
