# Peer review of "Occurrence of Antimicrobial Resistance Genes in the Oral Cavity of Cats with Chronic Gingivostomatitis"

_animals, 2021, doi:10.3390/ani11123589_

Round 1

Reviewer 1 Report

In the present study Authors measured the presence of antimicrobial resistance (AMR) genes in oral swab samples of 14 clinically healthy cats, and 14 sick cats with feline chronic gingivostomatitis (FCGS). After, sequencing 21 AMR genes were detected in healthy cats and 23 in FCGS. Later, it was demonstrated that AMR genes mecA and mphD had significant co-occurrence with fungus Malassezia restricta. In my opinion, study is well performed, and manuscript is nicely written, but it is not mentioned how 80 reference genes were selected? I suggest using established protocols to identify the AMR genes. For example: CARD (DOI: 10.1093/nar/gkw1004)  and PlasmidFinder (DOI: 10.1128/AAC.02412-14) can be integrated with  ARIBA (DOI: 10.1099/mgen.0.000131) to find out AMR genes (located both in genome and plasmid) in sequencing reads.

Author Response

We thank the reviewer for the input. Comments have been addressed in the body of the manuscript in the Material and Methods, section 2.3: Detection of AMR Genes, and Correlation Analysis with Malassezia Restricta; lines 121-134. Underlined text

Reviewer 2 Report

The article is well structured and reader friendly and the authors have demonstrated the presence of AMR genes in cats with oral disease per the aim of the study. References are relevant and up to date.
However, the reference to your previous work on Malassezia in cats with FCGS is interesting but circumstantial and irrelevant to this study. If you would like to include the finding that Malassezia co-occurs with AMR genes, please provide your methods to establishing this and reasoning behind the decision to test this particular species of fungus. You can also suggest a possible cause for this finding, based the known properties of fungi, especially commensal ones.  
Since FCGS is multifactorial, and viral infections are thought to play an important role, it might be useful to include the signalment of the cats that participated in the study, if relevant (for example co existing viral infections, outdoor/indoor and previous antibiotic therapy administrated). 

In the discussion, it might be beneficial to elaborate on the pathologic significance of these genes in causing disease. 

Line 43: the reference provided for this statement is incorrect, as the cited paper does not correlate with this statement. Please provide another refence and briefly describe the disease (example of reference: Murphy, Bell, Soukup. feline chronic gingivostomatitis. In veterinary oral and maxillofacial pathology.  willey Blackwell, 2020).

Line 48-50- reference needed.

Author Response

1) However, the reference to your previous work on Malassezia in cats with FCGS is interesting but circumstantial and irrelevant to this study. If you would like to include the finding that Malassezia co-occurs with AMR genes, please provide your methods to establishing this and reasoning behind the decision to test this particular species of fungus. 
Excellent point raised by the reviewer, a paragraph has been added to the body of the manuscript with regard to the rationale to test this yeast, as well as the unique properties of M restric Discussion lines 222 - 234. Underlined text

2) You can also suggest a possible cause for this finding, based the known properties of fungi, especially commensal ones.
This point has been addressed in the point 1 above

3) Since FCGS is multifactorial, and viral infections are thought to play an important role, it might be useful to include the signalment of the cats that participated in the study, if relevant (for example co existing viral infections, outdoor/indoor and previous antibiotic therapy administrated).
Information has been added to the manuscript in the section of Material and Methods, section 2.1 Subjects Included and Study Design, lines 75 – 98. Underlined text

4) In the discussion, it might be beneficial to elaborate on the pathologic significance of these genes in causing disease.
Points have been addressed in the manuscript. Lines 173 - 221

5) Line 43: the reference provided for this statement is incorrect, as the cited paper does not correlate with this statement. Please provide another refence and briefly describe the disease (example of reference: Murphy, Bell, Soukup. feline chronic gingivostomatitis. In veterinary oral and maxillofacial pathology.  willey Blackwell, 2020). 
Reference has been corrected – line 45

6) Line 48-50- reference needed.
Reference has been added – line 51

Round 2

Reviewer 2 Report

please consider showing some of the epidemiological data in a table. I think it will be easier for the reader. 

Author Response

Dear scientific reviewer,

We appreciate this comment, which is certainly a valid suggestion. However, this work was originally designed as an exploratory pilot study conducted with limited resources. As such, we collected some epidemiological data at the time of collection, but not consistently. The samples were obtained from multiple clinics and we neither had the personnel, nor the time/resources to collect comprehensive epidemiological data. Consequently, a table with epidemiological data would seem incomplete and would not add the expected value. Instead, we have modified the text in the manuscript to emphasize this is a pilot/descriptive study. Also, the data provides a foundation for a future prospective, randomized, clinical study that will include all relevant epidemiological data. We thank you again for your valuable input!